# Clinical Significance of the Lymph Node Ratio of the Second Operation to Predict Re-Recurrence in Thyroid Carcinoma

**DOI:** 10.3390/cancers15030624

**Published:** 2023-01-19

**Authors:** Joonseon Park, Il Ku Kang, Ja Seong Bae, Jeong Soo Kim, Kwangsoon Kim

**Affiliations:** Department of Surgery, College of Medicine, The Catholic University of Korea, Seoul 06591, Republic of Korea

**Keywords:** re-recurrence, lymph node ratio, papillary thyroid carcinoma, recurrence risk, mortality risk

## Abstract

**Simple Summary:**

Although the prognosis for thyroid cancer is favorable with a low mortality rate, the recurrence rate is high and predictors of re-recurrence and disease-specific mortality have not been established. This study revealed that multiple recurrences of thyroid cancer are associated with a higher mortality rate than a single recurrence, and the optimal treatment for the first recurrence is surgery. The lymph node ratio is a more suitable prognostic predictor than the number of positive lymph nodes in patients who undergo re-operation for the first recurrence of thyroid cancer.

**Abstract:**

The purpose of this study was to establish the risk factors for re-recurrences and disease-specific mortality (DSM) in recurrent thyroid cancer. Patients with recurrent thyroid cancer who underwent initial thyroid surgery from January 2000 to December 2019 at Seoul St. Mary’s Hospital (Seoul, Korea) were assessed. Clinicopathological characteristics and long-term oncologic outcomes were compared between patients with one recurrence (*n* = 202) and patients with re-recurrences (*n* = 44). Logistic regression and cox-regression analyses were conducted to determine the risk factors for re-recurrences and DSM, respectively. Receiver-operating characteristic curve analysis was performed to determine the cutoff value for lymph node ratio (LNR) as a predictor of re-recurrences. DSM was significantly higher in the re-recurrence group compared with the single-recurrence group (6.8% vs. 0.5%, *p* = 0.019). Surgical treatment at the first recurrence significantly lowered the risk of re-recurrences. Age (≥55), male sex, and LNR (≥0.15) were independent significant risk factors for re-recurrences in patients who underwent surgery at the first recurrence. Surgical resection is the optimal treatment for initial thyroid cancer recurrence. LNR at re-operation is more effective in predicting re-recurrence than the absolute number of metastatic LNs.

## 1. Introduction

Thyroid cancer is a slow-growing type of cancer with a good prognosis and a low disease-specific mortality (DSM) rate of less than 2% [1,2,3,4,5]. The DSM of thyroid cancer has been declining worldwide over the long term [6,7]. DSM prediction and active treatment to prevent thyroid cancer are essential. Although the DSM rate for thyroid cancer is low, tumor recurrence is common and the 10-year recurrence rate of papillary thyroid cancer (PTC) after initial surgery is approximately 15–20% [1,8,9,10]. Clinicopathological features, such as large tumor size, extrathyroidal extension, multifocality, metastatic lymph nodes (LNs), and local invasiveness, are risk factors for structural recurrence. The American Thyroid Association (ATA) management guidelines proposed a risk stratification system of structural disease recurrence based on these features [11]. Surgical resection is the optimal treatment for structural recurrence, and TSH–suppression, radioactive iodine (RAI) therapy, and kinase inhibitors can be employed after surgery if needed.

Re-recurrence refers to more than one recurrence. Various risk factors, including recurrence size, the number of recurrent LNs, and serum thyroglobulin (Tg) levels after the first re-operation, adversely affect re-recurrence-free survival (RFS) [12,13,14,15,16,17]. Surgical resection is the principal treatment for re-recurrence, as it is for the initial recurrence. However, ethanol ablation (EA) and radiofrequency ablation (RFA) can be used if surgery is challenging and non-surgical treatment is preferred in cases of repeated recurrences at the same site [18,19].

RFS based on the management strategy at the first recurrence has been studied [12,13,14]. However, due to the low mortality rate, few studies have evaluated DSM risk. In addition, the number of recurrent LNs is an independent risk factor for re-recurrence [12,20,21], but few studies have considered the lymph node ratio (LNR) based on the re-operation surgical outcome.

The prognostic value of LNR for predicting recurrence in thyroid cancer has been verified, but most studies predicted only the first recurrence according to the first surgical record [22,23,24]. Since the recurrent site, lesion volumes, and surgical extent vary compared to the initial surgery [20,21,25,26,27,28,29,30,31], the LNR according to the re-operation outcome may be more suitable for prognostic evaluation than the number of LNs. This study aims to evaluate the risk factors for re-recurrence and DSM in recurrent thyroid cancer and verify the significance of LNR as a predictor of re-recurrence.

## 2. Materials and Methods

### 2.1. Patients

We retrospectively reviewed 273 patients with recurrent thyroid cancer who underwent initial thyroid surgery between January 2000 and December 2019 at Seoul St. Mary’s Hospital (Seoul, Korea). Patients with distant metastasis at the initial operation and the first recurrence were excluded. Ten patients with distant metastasis at the first recurrence were excluded and four, seven, and six patients were excluded due to transfer, loss of follow-up, and no further treatment (observation only) due to the patient’s decision, respectively. Thus, data from the medical charts and pathology reports of 246 patients were analyzed. The mean follow-up duration was 129.1 ± 44.6 months (range: 33–264 months). This study was conducted in accordance with the Declaration of Helsinki (as revised in 2013) and approved by the Institutional Review Board of Seoul St. Mary’s Hospital, Catholic University of Korea (IRB No: KC22RISI0677), which waived the requirement for informed consent due to the retrospective nature of the study.

### 2.2. Follow-Up Assessment

The 2015 ATA management guidelines were followed for postoperative care and follow-up [11]. All patients were physically examined every 3–6 months for the first year followed by yearly exams. Serum thyroid function and Tg and anti-Tg antibody concentrations were measured. Thyroid ultrasonography was performed annually for the first five years, then biannually. RAI ablation was performed 10–12 weeks after total thyroidectomy, and whole-body scans were conducted 5–7 days after RAI ablation in patients with intermediate- and high-risk differentiated thyroid cancer.

### 2.3. Confirmation of Recurrence

Structural disease recurrence was confirmed based on the risk stratification of structural disease recurrence after initial therapy in the ATA management guidelines [11]. For biochemical incomplete response (BIR), recurrence was defined as structural recurrence confirmed through additional imaging and pathological examinations. This assessment was equally applied to re-recurrence and subsequent recurrences. RFS was defined as the period from the first recurrence to the second recurrence. During follow-up evaluations, patients with suspected recurrence underwent additional diagnostic imaging, including computed tomography, positron emission tomography/computed tomography, and RAI whole-body scan if recommended in intermediate- or high-risk patients following RAI therapy. After the precise location of the structural recurrence was detected and localized by diagnostic imaging, the recurrence was confirmed by pathologic diagnosis using ultrasound-guided fine-needle aspiration/core needle biopsy or surgical biopsy specimens.

### 2.4. Statistical Analysis

Continuous variables are presented as means with standard deviations, and categorical variables are reported as numbers with percentages. Clinicopathological characteristics were compared between the single-recurrence and re-recurrence groups. Continuous variables were compared with Student’s *t*-tests and categorical variables were compared using Pearson’s chi-square or Fisher’s exact tests. Factors associated with re-recurrence were determined using univariate and multivariate logistic regression analyses. Odds ratios (ORs) with 95% confidence intervals (CIs) were calculated using linear logistic regression analysis to compare re-recurrence risks for the independent factors. Disease-specific survival (DSS) predictors were identified by univariate cox-regression analyses, and statistically significant variables were included in the multivariate Cox proportional hazard model. Hazard ratios (HRs) with 95% CIs were calculated. The cutoff value for LNR for predicting re-recurrence was determined by calculating the area under the curve (AUC) from receiver-operating characteristic (ROC) curve analysis. RFS and DSS were assessed using Kaplan–Meier survival analysis, and the log-rank test was used to calculate significant differences. Differences with *p*-values of less than 0.05 were considered statistically significant. All statistical analyses were performed using Statistical Package for the Social Sciences (version 24.0; IBM Corp., Armonk, NY, USA).

## 3. Results

### 3.1. Comparison of Clinicopathological Characteristics According to Re-Recurrence

The clinicopathological characteristics of all patients are shown in Table 1. Of the 246 patients included in the study, 169 (68.7%) were female. The mean patient age was 43.9 ± 14.3 (range: 11–74). The single-recurrence group included 202 patients and the re-recurrence group included 44 patients. The mean age was significantly older for the re-recurrence group compared to that of the single-recurrence group (48.4 ± 16.5 vs. 42.9 ± 13.6, *p* = 0.044). The surgery was significantly more extensive in the re-recurrence group (*p* < 0.001), and the ratio of medullary thyroid carcinoma (MTC) or poorly differentiated thyroid carcinoma (PDTC)/anaplastic thyroid (ATC) was significantly higher in the re-recurrence group compared to the cancer type of the single-recurrence group (*p* = 0.003). The re-recurrence group had significantly more positive and harvested LNs than the single-recurrence group (*p* = 0.031, *p* = 0.013, respectively). T category, N category, and TNM category were significantly more advanced in the re-recurrence group than in the single-recurrence group *(p* < 0.001, *p* < 0.001, *p* = 0.003, respectively). The rate of surgical treatment (surgery ± RAI) at the first recurrence was significantly lower in the re-recurrence group than in the single-recurrence group (93.1% vs. 81.8%, *p* = 0.036). Both overall mortality (OM) and DSM were significantly higher in the re-recurrence group than in the single-recurrence group (OM: 11.4% vs. 1.5%, *p* = 0.005; DSM: 6.8% vs. 0.5%, *p* = 0.019).

### 3.2. Logistic Regression Analysis of Risk Factors for Re-Recurrence Based on First Operation Outcomes

Table 2 shows the univariate and multivariate logistic regression analyses of risk factors for the re-recurrence of thyroid cancer based on the results of the initial surgery. The univariate analysis shows that age (≥55), male sex, PDTC/ATC, tumor size, gETE, vascular invasion, the number of positive LNs, T category, N category, TNM stage, and management at the first recurrence were significant risk factors for re-recurrence. According to the multivariate analysis, vascular invasion (OR, 4.348; 95% CI, 1.247–15.166; *p* = 0.021), T3 category (OR, 2.938; 95% CI, 1.124–7.680; *p* = 0.028), and RFA rather than surgery (OR, 4.249; 95% CI, 1.095–16.495; *p* = 0.037) were significant risk factors for re-recurrence. As shown in Figure 1, RFS was significantly different according to management strategies at the first recurrence, and surgical treatment showed the highest RFS (*p* = 0.045).

### 3.3. Cox-Regression Analysis of Risk Factors for DSM in Patients with Recurrent Thyroid Cancer

The results of the univariate and multivariate cox-regression analyses to identify significant risk factors for DSM are shown in Table 3. According to the univariate analysis, age, PDTC/ATC, tumor size, TNM stage, and management at the first recurrence were significant risk factors for DSM. In the multivariate analysis, management at the first recurrence was the only significant risk factor for DSM. As shown in Figure 2, DSS was significantly different for different management strategies at the first recurrence; surgical treatment showed the best RFS (*p* = 0.002).

### 3.4. Surgical Outcomes of Re-Operation at the First Recurrence

For patients who underwent re-operation at the first recurrence, a comparison analysis was conducted by dividing the single recurrence group and the re-recurrence group according to the results of the re-operation (Table 4). The single-recurrence and re-recurrence groups consisted of 188 and 36 patients, respectively. In the single-recurrence group, the first recurrence was significantly more frequent in the contralateral lobe, and in the re-recurrence group, the first recurrence was significantly more frequent in the central compartment, lateral compartment, and distant metastasis (*p* = 0.003). The ratio of MTC and PDTC/ATC in the re-recurrence group was significantly higher than the cancer type ratio in the single recurrence group (MTC: 2.8% vs. 1.1%; PDTC/ATC: 8.3% vs. 1.1%, respectively, *p* = 0.018). LNR at re-operation was significantly higher in the re-recurrence group compared with the LNR in the single recurrence group (0.42 ± 0.38 vs. 0.22 ± 0.27, *p* = 0.005). Both OM and DSM were significantly higher in the re-recurrence group than in the single recurrence group (OM: 11.1% vs. 1.1%, *p* = 0.007; DSM: 5.6% vs. 0.0%, *p* = 0.025, respectively).

### 3.5. Optimal Cutoff Values

The results of the ROC curve analysis for LNR and positive LNs are shown in Figure 3. The AUC for LNR was better than the AUC for the number of positive LNs (AUC, 0.658 vs. 0.489; *p*-value, 0.003 vs. 0.829). The optimal cutoff value for LNR was 0.15 (sensitivity, 0.61; specificity, 0.61).

### 3.6. Logistic Regression Analysis of Risk Factors for Re-Recurrence Based on Re-Operation Outcomes after the First Recurrence

Table 5 shows the risk factors for re-recurrence based on the re-operation outcomes. In the multivariate analysis, age (≥55) and male sex were independent risk factors for re-recurrence. Within the surgical outcomes, only LNR (cutoff 0.15) was a significant risk factor for re-recurrence (OR, 2.536; 95% CI, 1.181–5.446; *p* = 0.017). Figure 4 shows the RFS according to the LNR with a cutoff of 0.15. RFS was significantly lower in patients with LNR ≥ 0.15 compared with RFS in patients with LNR < 0.15.

## 4. Discussion

Our results demonstrate that DSM and RFS are significantly different for different management strategies at the time of the primary recurrence, and surgery is the optimal treatment. In addition, the LNR at re-operation predicted re-recurrence better than the absolute number of metastatic LNs, and the cutoff value of LNR was 0.15 for predicting re-recurrence.

The 10-year recurrence rate for thyroid cancer ranges from 5% to 35% [13,24,28,29]. Recurrence rates vary according to clinicopathological characteristics. Thus, the ATA guidelines suggest a risk stratification system based on the different risk factors. The recurrence rate is approximately 1–5% in the low-risk group, 5–20% in the intermediate-risk group, and up to 55% in the high-risk group, showing more specific and continuous stratification according to each feature [11].

The present study included a cohort of patients with recurrent thyroid cancer, with a mean follow-up period of approximately 129.1 months. During the 10-year follow-up period, re-recurrences were observed in 44 (18%) patients. This result is consistent with prior studies estimating a 5-year RFS rate of approximately 75–85% [12,14]. Xu et al. found that the 5- and 10-year RFS rates were 85.5% and 79.9%, respectively, after re-operation for the initial recurrence of thyroid cancer. In addition, the 5-year RFS rates according to the ATA risk stratification were 92.4%, 86.2%, and 74.5% in the low, intermediate, and high-risk groups, respectively, with no significant differences in adjacent groups (all *p* > 0.05) [12].

The most common recurrence sites in this study were similar to those reported in previous studies. Lee et al. found that when hemithyroidectomy was performed in the initial operation, the most common recurrence sites were the contralateral lobe and central compartment, followed by the lateral compartment. After total thyroidectomy, the initial recurrences were in the lateral compartment, central compartment, and remnant thyroid (67.5, 31.8, and 18.5%, respectively). Re-recurrences were found most commonly in the central compartment, followed by the lateral compartment and distant metastasis [14]. We analyzed recurrence sites regardless of the extent of the initial operation. Similar to the previous studies, initial recurrence sites, in order of incidence, were the lateral compartment, central compartment, contralateral lobe, and systemic metastases. Re-recurrence sites, in order of incidence, were the lateral compartment, central compartment, and systemic metastasis. Patients with remnant thyroid underwent complete thyroidectomy at the initial recurrence; thus, the frequency of re-recurred sites was similar to that of the initial recurrence, considering that the contralateral lobe was not left. The most frequently involved organ for both the initial recurrence and re-recurrences was LN, and the primary treatment was surgery.

The present study supports previous research showing that surgical removal of the recurrent lesion is the optimal strategy for treating recurrent lesions [16,17]. Osama et al. demonstrated no mortality and relatively low morbidity when surgical resection of recurrent or persistent RAI-resistant PTC was performed by experts, and biochemical complete remission (BCR) was achieved in 27% of patients who had not been treated with other methods [16]. Long-term follow-up (approximately 13 years) of the same cohort showed that surgical resection of recurrent or persistent PTC in cervical LNs resulted in a BCR of 25.7% [17]. RFA, EA, and RAI are alternatives to surgery for patients with inoperable lesions or those who do not wish to undergo repeated surgery [18,19]. Monchik et al. demonstrated that 1 out of 13 patients (7.7%) with RFA had a re-recurrence near the treated site, and that patient underwent a modified neck dissection, with the developing disease being treated by RFA [32]. Similarly, Baek et al. verified the therapeutic success of RFA in 10 out of 12 tumors (83%) and 1 in 10 patients (10%) experienced re-recurrences [33]. Kim et al. demonstrated that RFA performed in an outpatient clinic without general anesthesia was beneficial for treating tumors smaller than 2 cm [19]. Moreover, patients sometimes reject repeated reoperations, but RFA should be performed with caution due to complications such as hoarseness caused by recurrent laryngeal nerve injury or skin burns [32].

RAI performance or dose did not affect DSM or re-recurrence in this study. Several studies suggest that RAI therapy after re-operation has no significant benefit in preventing re-recurrences [14,34,35]. Yim et al. found that adjuvant RAI therapy had no significant effects on stimulated Tg changes or RFS compared with no additional RAI therapy [34]. Although adjuvant RAI may reduce the risk of re-recurrence, differences in the effects of RAI therapy compared with re-operation in patients with a single recurrence are unclear.

In the present study, 36 out of 244 patients (23%) in the surgical group, 5 out of 16 patients (31%) in the RFA group, and 3 out of 6 patients (50%) in the systemic therapy group (RAI or TKI) experienced re-recurrences. In the multivariate analysis, the re-recurrence risk was higher in patients who underwent RFA compared with those who underwent surgery, with an OR of 4.249. In addition, the only risk factor for DSM was the treatment strategy at the initial recurrence, and surgery was the optimal strategy. RFA or RAI may not eradicate gross tumors in the thyroid bed. Decisions concerning treatment strategies for recurrent thyroid cancer are often difficult in complicated cases, and making collaborative decisions between the patient and a multidisciplinary clinical team is best [17]. At our institution, the department of endocrinology, pathology, radiology, otolaryngology, nuclear medicine, surgery, and oncology form a multidisciplinary team, and the optimal treatment is determined after discussing appropriate strategies with recurrent patients. Except for inoperable cases, surgical resection is usually the preferred strategy.

This study included patients with structural recurrences after the initial surgery, and the re-recurrence or mortality risk assessment may differ in patients initially diagnosed with thyroid cancer. ATA management guidelines strongly recommend continually adjusting recurrence risk stratification throughout the follow-up period as initial staging systems provide static, single-point estimations based only on data at the time of initial treatment. The risk of re-recurrence and DSM may change over time depending on the clinical course of the illness and the response to treatment [11]. For example, aggressive histology, such as PTC variants, predicts DFS at initial operation [36,37,38,39]. However, our results show that LNR and the relative volume of malignancy are better prognostic factors than aggressive variants of PTC in patients with recurrences. Therefore, the surgical outcome after re-operation may play an important role in the risk of re-recurrence.

Lamartina et al. found that age (≥45), aggressive histology, and LNR (>0.6) at the initial surgery were risk factors for BIR or structural incomplete responses after the first re-operation (*p* = 0.005, *p* = 0.005, and *p* = 0.009, respectively). Independent risk factors for re-recurrence after complete remission following the initial re-operation were male sex (HR, 5.2; *p* = 0.006), aggressive histology (HR, 10.3; *p* = 0.009), and the number of positive LNs >10 at re-operation (HR, 6.7; *p* = 0.01) [20]. Unlike our study, aggressive histology and more than 10 positive LNs but not LNR were independent risk factors for re-recurrence.

LNR is the ratio of the number of metastatic LNs divided by the total number of harvested LNs. Since it is a relative ratio, the metastatic volume can be adequately estimated regardless of the number of harvested LNs. The prognostic significance of LNR has been studied in various cancers, including breast, thyroid, colon, and stomach cancers [22,23,24,40,41,42,43,44]. In gastric cancer, LNR is influenced by the extent of surgery and the disease burden. At least 16 LNs are required to evaluate the N category in gastric cancer; thus, many studies have shown that the LNR is a better predictor of mortality than the N category in high-risk or elderly patients who need limited LN dissection [42,45]. Moreover, Wang et al. verified that LNR is a more accurate predictor of mortality than pN stage in stage III colon cancer [40]. In head and neck cancer, the LNR was superior to the traditional pN stage in predicting poor prognosis and the need for adjuvant therapy [46]. In recurrent thyroid cancer, the LNR may estimate outcomes better than the total number of positive LNs as the LNR considers the extent of surgery [23]. Yu et al. demonstrated that an optimal evaluation of positive LNs can be performed with at least 11 or more harvested LNs, and this supports the rationale for LNR when the number of harvested LNs is low [47]. Lamartina et al. showed that LNRs of primary surgery and re-operation were not independent risk factors for re-recurrence in patients with DTC [20]; however, median values were used to select the cutoff value for LNR rather than the ROC curve in that study. In our study, the cutoff LNR value of 0.15 was determined based on the ROC curve analysis, and an LNR higher than 0.15 may predict re-recurrence. Numerous studies that demonstrate the impact of LNR as a predictor used ROC curves to determine the cutoff value, similar to the present study [40,43,44].

There are several limitations to this study. First, this was a retrospective, single-center study. Second, postoperative Tg values and BIR were not analyzed. Previous studies showed that stimulated Tg is a significant predictor of re-recurrence after the initial re-operation in patients with recurrent PTC [14,15,16]. Most studies defined BIR as stimulated Tg ≥ 1 ng/mL [14,15,16], and Lee et al. demonstrated that BIR significantly lowered RFS after 5 years compared to BCR (HR, 3.191; 95% CI, 1.519–6.705; *p* = 0.001) [14]. In our study, if BIR was suspected, only cases with a structural incomplete response diagnosed with additional imaging and pathologic confirmation were included. We will investigate more developed outcomes by including both unstimulated and stimulated postoperative Tg values in the future. Third, although we followed up for an average of 129 months, we found very few cases of DSM. Therefore, a multicenter study with a larger sample size that includes information such as postoperative Tg values and biochemical responses could support the reliability of this study.

## 5. Conclusions

The optimal treatment for the initial recurrence in patients with thyroid cancer is surgical resection with a low risk of DSM and better RFS. The LNR at re-operation is more effective in predicting re-recurrence than the absolute number of metastatic LNs. An LNR ≥ 0.15 is an independent predictor of RFS and careful follow-up is needed.

## Figures and Tables

**Figure 1 cancers-15-00624-f001:**
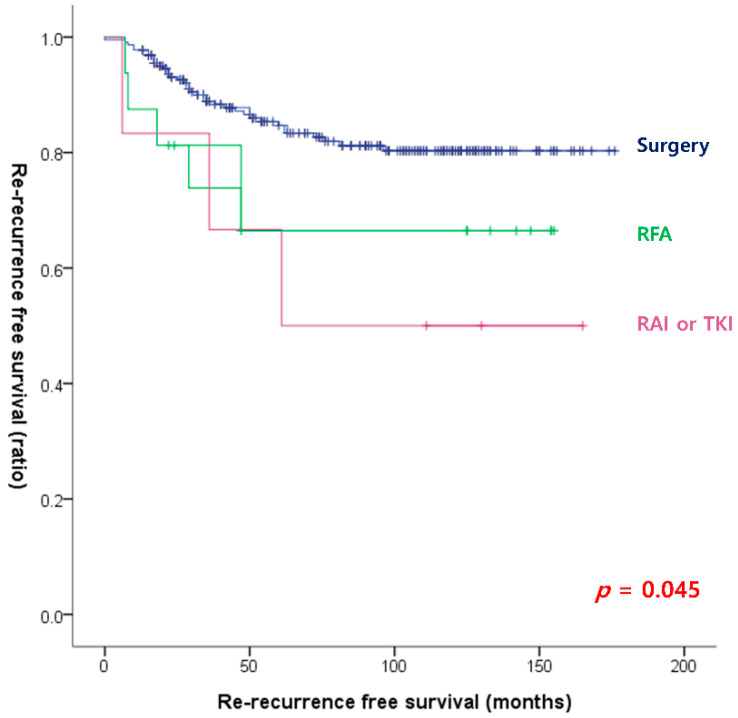
Re-recurrence-free survival curves according to the management at the first recurrence (log-rank *p* = 0.045).

**Figure 2 cancers-15-00624-f002:**
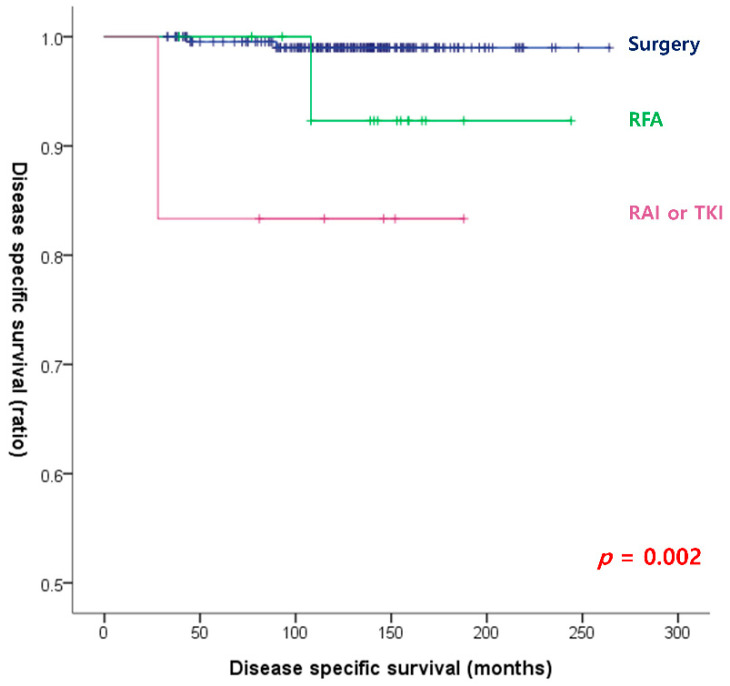
Disease-specific survival curves according to the management at the first recurrence (log-rank *p* = 0.002).

**Figure 3 cancers-15-00624-f003:**
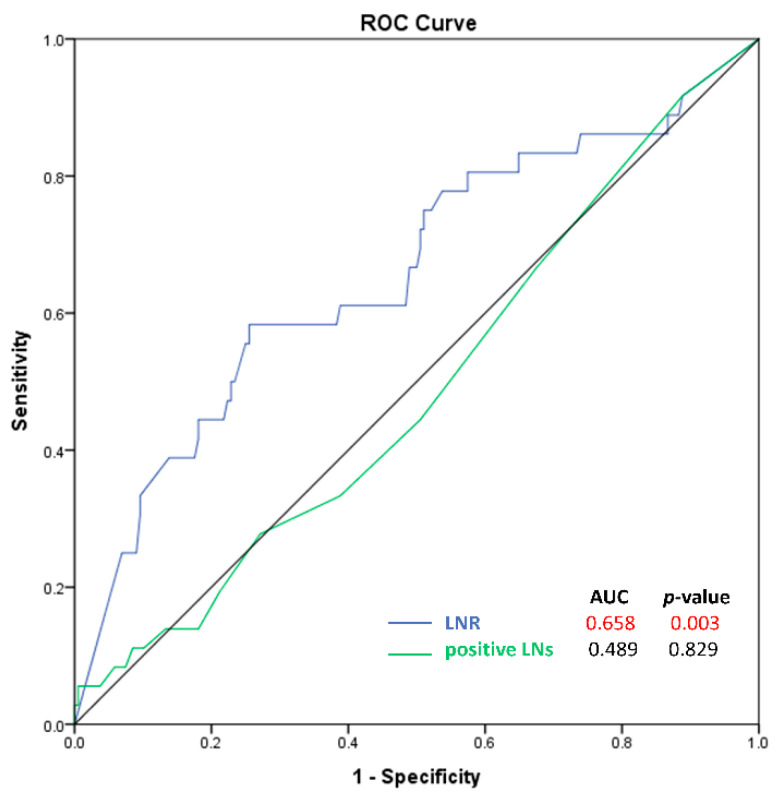
Receiver-operator characteristic curves for lymph node ratio and the number of positive lymph nodes.

**Figure 4 cancers-15-00624-f004:**
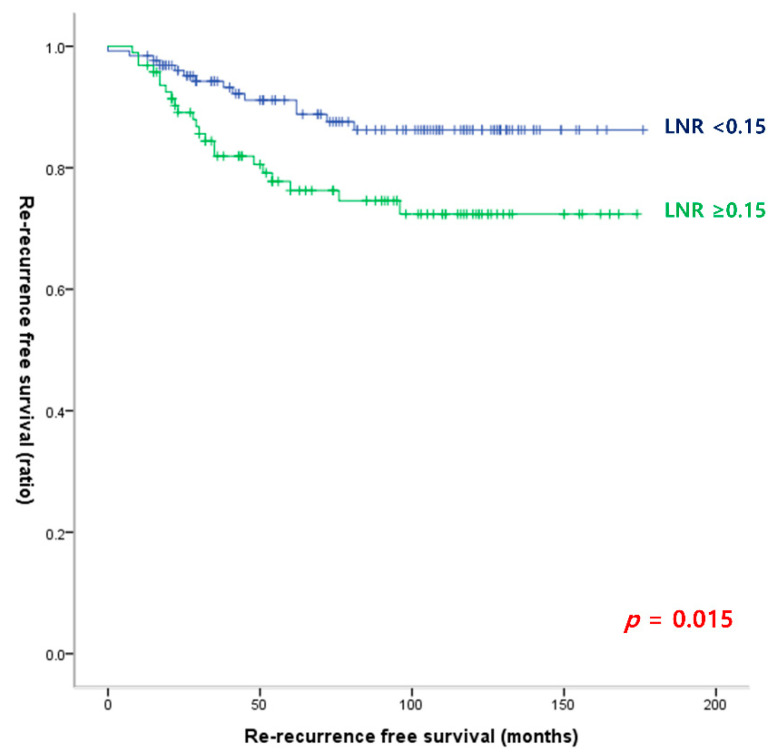
Re-recurrence-free survival curves according to the lymph node ratio cutoff of 0.15 (log-rank *p* = 0.015).

**Table 1 cancers-15-00624-t001:** Comparison of clinicopathological characteristics according to re-recurrence.

	Recurrence (*n* = 246)	Single-Recurrence (*n* = 202) (A)	Re-Recurrence (*n* = 44) (B)	*p*-Value (A vs. B)
Age (years)	43.9 ± 14.3 (range, 11–74)	42.9 ± 13.6 (range, 11–74)	48.4 ± 16.5 (range, 12–72)	0.044
Age (≥55)	58 (23.6%)	41 (20.3%)	17 (38.6%)	0.009
Female	169 (68.7%)	145 (71.8%)	24 (54.5%)	0.025
Obesity (BMI ≥ 25 kg/m^2^)	91 (37.0%)	74 (36.6%)	17 (38.6%)	0.803
Surgical extent				<0.001
Less than total	56 (22.8%)	53 (26.2%)	3 (6.8%)	
Total thyroidectomy	125 (50.8%)	105 (52.0%)	20 (45.5%)	
Lateral neck dissection	65 (26.4%)	44 (21.8%)	21 (47.7%)	
Cancer type				
PTC/FTC	235 (95.5%)	197 (97.5%)	38 (86.4%)	0.003
MTC	5 (2.0%)	3 (1.5%)	2 (4.5%)	
PDTC/ATC	6 (2.4%)	2 (1.0%)	4 (9.1%)	
PTC variant				0.479
Non-aggressive	208/225 (92.4%)	176/192 (91.7%)	32/33 (97.0%)	
Aggressive	17/225 (7.6%)	16/192 (8.3%)	1/33 (3.0%)	
Tumor size (cm)	1.7 ± 1.3 (range, 0.2–8.0)	1.5 ± 1.2 (range, 0.2–6.5)	2.5 ± 1.7 (range, 0.2–8.0)	0.001
gETE	37 (15.0%)	23 (11.4%)	14 (31.8%)	0.001
Multifocality	120 (48.8%)	100 (49.5%)	20 (45.5%)	0.626
Bilaterality	78 (31.7%)	62 (30.7%)	16 (36.4%)	0.464
Vascular invasion	18/231 (7.8%)	9 (4.7%)	9 (22.0%)	0.001
Lymphatic invasion	140/236 (59.3%)	114/195 (58.5%)	26/41 (63.4%)	0.557
Perineural invasion	14/232 (6.0%)	11/191 (5.8%)	3/41 (7.3%)	0.718
BRAF positive	175/212 (82.5%)	150/180 (83.3%)	25/32 (78.1%)	0.474
LNR	0.37 ± 0.30 (range, 0.0–1.0)	0.37 ± 0.30 (range, 0.0–1.0)	0.41 ± 0.31 (range, 0.0–1.0)	0.500
Positive LNs	7.5 ± 9.6 (range, 0–74)	6.6 ± 7.8 (range, 0–45)	12.4 ± 15.5 (range, 0–74)	0.031
Harvested LNs	22.9 ± 27.7 (range, 1–185)	20.4 ± 25.6 (range, 1–185)	35.8 ± 34.5 (range, 1–132)	0.013
T category				<0.001
T1	164 (66.7%)	146 (72.3%)	18 (40.9%)	
T2	35 (14.2%)	26 (12.9%)	9 (20.5%)	
T3a	46 (18.7%)	7 (3.5%)	4 (9.1%)	
T3b		23 (11.4%)	12 (27.3%)	
T4a	1 (0.4%)	0 (0.0%)	1 (2.3%)	
N category				<0.001
Nx	17 (6.9%)	10 (5.0%)	7 (15.9%)	
N0	39 (15.9%)	36 (17.8%)	3 (6.8%)	
N1a	125 (50.8%)	111 (55.0%)	14 (31.8%)	
N1b	65 (26.4%)	45 (22.3%)	20 (45.5%)	
TNM stage				0.003
Stage I	201 (81.7%)	172 (85.1%)	29 (65.9%)	
Stage II	45 (18.3%)	30 (14.9%)	15 (34.1%)	
RAI ablation	185 (75.2%)	147 (72.8%)	38 (86.4%)	0.059
RAI dose	151.5 ± 85.5 (range, 0–700)	146.9 ± 84.1 (range, 0–700)	168.5 ± 89.6 (range, 0–450)	0.157
1st recur site				0.010
Contralateral lobe	20 (8.1%)	20 (9.9%)	0 (0.0%)	
Central compartment	43 (17.5%)	35 (17.3%)	8 (18.2%)	
Lateral compartment	173 (70.3%)	142 (70.3%)	31 (70.5%)	
Systemic metastasis	10 (4.1%)	5 (2.5%)	5 (11.4%)	
1st recur management				0.036
Surgery ± RAI	224 (91.1%)	188 (93.1%)	36 (81.8%)	
RFA ± RAI	16 (6.5%)	11 (5.4%)	5 (11.4%)	
Systemic therapy only (RAI or TKI)	6 (2.4%)	3 (1.5%)	3 (6.8%)	
Variant change	10/239 (4.2%)	7/197 (3.6%)	3/37 (8.1%)	0.198
Mortality (Overall)	8 (3.3%)	3 (1.5%)	5 (11.4%)	0.005
Disease-specific Mortality	4 (1.6%)	1 (0.5%)	3 (6.8%)	0.019

Data are expressed as the patient’s number (%) or the mean ± standard deviation. A statistically significant difference was defined as *p* < 0.05. Abbreviations: PTC, papillary thyroid cancer; FTC, follicular thyroid cancer; MTC, medullary thyroid cancer; PDTC, poorly differentiated thyroid cancer; ATC, anaplastic thyroid cancer; gETE, gross extrathyroidal extension; BRAF, B-Raf proto-oncogene; LNR, lymph node ratio; LN, lymph node; T, tumor; N, node; TNM, tumor-node-metastasis; RAI, radioactive iodine; RFA, radiofrequency ablation; TKI, tyrosine kinase inhibitor.

**Table 2 cancers-15-00624-t002:** Logistic regression analysis of risk factors for re-recurrence according to 1st op outcomes.

	Univariate		Multivariate	
	OR (95% CI)	*p*-Value	OR (95% CI)	*p*-Value
Age (≥55)	2.472 (1.231–4.964)	0.011		
Male	2.120 (1.087–4.134)	0.027		
Cancer type				
PTC/FTC	Ref.	0.014		
MTC	3.456 (0.559–21.386)	0.182		
PDTC/ATC	10.368 (1.834–58.633)	0.008		
Aggressive variant	0.344 (0.044–2.684)	0.308		
Tumor size (cm)	1.562 (1.250–1.952)	<0.001		
gETE	3.632 (1.684–7.834)	0.001		
Vascular invasion	5.656 (2.086–15.336)	0.001	4.348 (1.247–15.166)	0.021
Positive LNs	1.051 (1.017–1.086)	0.003		
T category				
T1	Ref.	0.002	Ref.	0.089
T2	2.808 (1.139–6.922)	0.025	1.604 (0.501–5.136)	0.426
T3	4.326 (1.984–9.434)	<0.001	2.938 (1.124-7.680)	0.028
T4	1.31 × 10^10^ (0.000–10^10^)	1.000		
N category				
N0	Ref.	<0.001		
N1a	1.514 (0.412–5.567)	0.533		
N1b	5.333 (1.468–19.379)	0.011		
Nx	8.400 (1.831–38.530)	0.006		
TNM stage				
Stage I	Ref.			
Stage II	2.966 (1.423–6.179)	0.004		
1st recurrence management				
Surgery ± RAI	Ref.	0.054	Ref.	0.080
RFA ± RAI	2.374 (0.778–7.244)	0.129	4.249 (1.095–16.495)	0.037
Systemic therapy only (RAI or TKI)	5.222 (1.013–26.909)	0.048	2.804 (0.382–20.578)	0.311

A statistically significant difference was defined as *p* < 0.05. Abbreviations: PTC, papillary thyroid cancer; FTC, follicular thyroid cancer; MTC, medullary thyroid cancer; PDTC, poorly differentiated thyroid cancer; ATC, anaplastic thyroid cancer; gETE, gross extrathyroidal extension; LN, lymph node; T, tumor; N, node; TNM, tumor-node-metastasis; RAI, radioactive iodine; RFA, radiofrequency ablation; TKI, tyrosine kinase inhibitor.

**Table 3 cancers-15-00624-t003:** Cox regression analysis of risk factors for mortality in patients with recurrent thyroid cancer.

	Univariate		Multivariate	
	HR (95% CI)	*p*-Value	HR (95% CI)	*p*-Value
Age (≥55)	9.940 (1.033–95.627)	0.047		
Male	2.349 (0.330–16.704)	0.393		
Cancer type				
PTC/FTC	Ref.	0.059	Ref.	0.993
MTC	0.000 (0.000–10^∞^)	0.992	0.000 (0.000–9.404 × 10^93^)	0.945
PDTC/ATC	15.910 (1.625–155.757)	0.017	0.867 (0.047–15.998)	0.924
Tumor size (cm)	1.982 (1.313–2.990)	0.001	2.005 (0.969–4.147)	0.061
Vascular invasion	5.438 (0.493–60.032)	0.167		
Positive LNs	0.870 (0.667–1.136)	0.306		
TNM stage				
Stage I	Ref.			
Stage II	14.148 (1.469–136.295)	0.022		
1st recurrence management				
Surgery ± RAI	Ref.	0.079	Ref.	0.043
RFA ± RAI	6.694 (0.607–73.861)	0.121	4.465 (0.315–63.333)	0.269
Systemic therapy only (RAI or TKI)	20.654 (1.870–228.080)	0.013	38.656 (1.810–825.392)	0.019

A statistically significant difference was defined as *p* < 0.05. Abbreviations: PTC, papillary thyroid cancer; FTC, follicular thyroid cancer; MTC, medullary thyroid cancer; PDTC, poorly differentiated thyroid cancer; ATC, anaplastic thyroid cancer; gETE, gross extrathyroidal extension; LN, lymph node; T, tumor; N, node; TNM, tumor-node-metastasis; RAI, radioactive iodine; RFA, radiofrequency ablation; TKI, tyrosine kinase inhibitor.

**Table 4 cancers-15-00624-t004:** Surgical outcomes in patients who underwent reoperation at the 1st recurrence.

	Recurrence (224)	Single-Recurrence (*n* = 188) (C)	Re-Recurrence (*n* = 36) (D)	*p*-Value (C vs. D)
1st recur site				0.003
Contralateral lobe	20 (8.9%)	20 (10.6%)	0 (0.0%)	
Central compartment	36 (16.1%)	30 (16.0%)	6 (16.7%)	
Lateral compartment	164 (73.2%)	137 (72.9%)	27 (75.0%)	
Systemic metastasis	4 (1.8%)	1 (0.5%)	3 (8.3%)	
Cancer type				0.018
PTC/FTC	216 (96.4%)	184 (97.9%)	32 (88.9%)	
MTC	3 (1.3%)	2 (1.1%)	1 (2.8%)	
PDTC/ATC	5 (2.2%)	2 (1.1%)	3 (8.3%)	
PTC variant at re-op				0.753
Non-aggressive	184 (88.5%)	159 (88.8%)	25 (86.2%)	
Aggressive	24 (4.2%)	20 (11.2%)	4 (13.8%)	
Change of PTC variant (to aggressive variant)	10/215 (4.7%)	7/184 (3.8%)	3/31 (9.7%)	0.161
Positive LNs at re-op	3.9 ± 4.6 (range: 0–45)	3.7 ± 3.7 (range: 0–20)	4.6 ± 7.9 (range: 0–45)	0.516
Harvested LNs at re-op	26.4 ± 21.1 (range: 1–124)	27.4 ± 20.1 (range: 1–73)	21.2 ± 25.1 (range: 1–124)	0.109
LNR at 2nd op	0.25 ± 0.30 (range: 0–1)	0.22 ± 0.27 (range: 0–1)	0.42 ± 0.38 (range: 0–1)	0.005
Positive LNs > 5	47 (21.0%)	40 (21.3%)	7 (19.4%)	0.805
RAI after re-op	102 (45.5%)	83 (44.1%)	19 (52.8%)	0.341
Mortality (Overall)	6 (2.7%)	2 (1.1%)	4 (11.1%)	0.007
Disease-specific Mortality	2 (0.9%)	0 (0.0%)	2 (5.6%)	0.025

Data are expressed as the patient’s number (%) or the mean ± standard deviation. A statistically significant difference was defined as *p* < 0.05. Abbreviations: PTC, papillary thyroid cancer; FTC, follicular thyroid cancer; MTC, medullary thyroid cancer; PDTC, poorly differentiated thyroid cancer; ATC, anaplastic thyroid cancer; LN, lymph node; LNR, lymph node ratio; RAI, radioactive iodine.

**Table 5 cancers-15-00624-t005:** Logistic regression analysis of risk factors for re-recurrence according to reoperation outcomes at the 1st recurrence.

	Univariate		Multivariate	
	HR (95% CI)	*p*-Value	HR (95% CI)	*p*-Value
Age (≥55)	2.915 (1.372–6.195)	0.005	3.147 (1.433–6.914)	0.004
Male	2.404 (1.159–4.983)	0.018	2.141 (1.004–4.566)	0.049
Cancer type				
PTC/FTC	Ref.	0.051	Ref.	0.387
MTC	2.875 (0.253–32.645)	0.394	0.923 (0.072–11.743)	0.950
PDTC/ATC	8.625 (1.386–53.668)	0.021	3.976 (0.552–28.660)	0.171
Aggressive variant at re-op	1.272 (0.401–4.031)	0.683		
Change of PTC variant (to aggressive variant)	2.709 (0.661–11.097)	0.166		
Positive LNs	1.035 (0.969–1.106)	0.310		
Harvested LNs	0.985 (0.967–1.003)	0.110		
LNR 0.15	2.476 (1.191–5.144)	0.015	2.536 (1.181−5.446)	0.017
Positive LNs >5	0.893 (0.364–2.188)	0.805		
RAI after re-op	1.414 (0.692–2.890)	0.342		

A statistically significant difference was defined as *p* < 0.05. Abbreviations: PTC, papillary thyroid cancer; FTC, follicular thyroid cancer; MTC, medullary thyroid cancer; PDTC, poorly differentiated thyroid cancer; ATC, anaplastic thyroid cancer; LN, lymph node; LNR, lymph node ratio; RAI, radioactive iodine.

## Data Availability

The data that support the findings of this study are available upon request from the corresponding author. The data are not publicly available due to privacy or ethical restrictions.

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
