# Peer review of "Clinical Significance of the Lymph Node Ratio of the Second Operation to Predict Re-Recurrence in Thyroid Carcinoma"

_cancers, 2023, doi:10.3390/cancers15030624_

Round 1
Reviewer 1 Report
The purpose of this study was to establish the risk factors for re-recurrences and disease- specific mortality (DSM) in recurrent thyroid cancer. Age (≥55), male sex, and LNR (≥0.15) were independent significant risk factors for re-recurrences in patients who underwent surgery at the first recurrence. Surgical resection is the optimal treatment for initial thyroid cancer recurrence. LNR at re-operation is more effective in predicting re-recurrence than the absolute number of metastatic LNs.
Several questions should be addressed.
1. Harvested LN is associated with the recurrence of PTC in the initial surgery. What about the LN distributed in central neck and lateral neck? what about their association with the prognosis?
2.In figure 3, authors conducted ROC curves for LNR and positive LNs for optimal cut-off values. However, I did not see the p-value?
3. Some papers recommended for authors: PMID: 34710736;PMID: 26825749;PMID: 31854077.
Author Response
- Harvested LN is associated with the recurrence of PTC in the initial surgery. What about the LN distributed in central neck and lateral neck? what about their association with the prognosis?
Response) In the present study, as surgical extent in Table 1, lateral neck dissection was performed significantly more in re-recurrence group than in the once-recurrence group (47.7% vs 21.8%, p<0.001). All other patients underwent prophylactic or curative central neck dissection in our study. Since lateral neck dissection was performed when lateral neck metastasis was confirmed preoperatively, lateral neck dissection at the initial operation might represent more aggressive and advanced initial stage. This would have resulted in a higher re-recurrence rate.
- In figure 3, authors conducted ROC curves for LNR and positive LNs for optimal cut-off values. However, I did not see the p-value?
Response) p- value of LNR is 0.003 and p- value of positive LNs is0.829, respectively. I added these values in figure 3 and ‘results’ in manuscript. We appreciate for pointing out what we missed.
- Some papers recommended for authors: PMID: 34710736; PMID: 26825749;PMID: 31854077.
Response) I cited the paper you recommended in ‘discussion’:
Yu et al. demonstrated that an optimal evaluation of positive LNs can be performed with at least 11 or more harvested LNs, and this supports the rationale for LNR when the number of harvested LNs is low [47].
We sincerely appreciate your comments that make our work improve further.
Thank you.
We thank you and the reviewers for the insightful comments. We believe that our manuscript has been improved as a direct result of the review process. We hope that the revised manuscript is now suitable for publication in CANCERS.
Sincerely,
Joonseon Park, MD
Kwangsoon Kim, MD, PhD

Reviewer 2 Report
Introduction:
LNR is subjective according to how much the surgeon decided to remove LN. Therefore, rationale for using this tool needs to be more addressed.
Tables2 and 3:
The multivariate cox regression model contains variables that ruin the output and should be removed. These very wide confidence interval and poor precision indicates lower insufficient sample size per group.
Author Response
LNR is subjective according to how much the surgeon decided to remove LN. Therefore, rationale for using this tool needs to be more addressed.
Response) Since LNR is a relative ratio, we considered that metastatic volume can be more adequately reflected by LNR than the number of positive LNs, regardless of the number of harvested LNs. In addition, it has been demonstrated in previous studies that the LNR is a reasonable estimate reflecting the surgical extent, and we mentioned these in ‘discussion’ as below:
LNR is the ratio of the number of metastatic LNs divided by the total number of harvested LNs. Since it is a relative ratio, the metastatic volume can be adequately estimated regardless of the number of harvested LNs ……….………In gastric cancer, LNR is influenced by the extent of surgery and the disease burden. At least 16 LNs are required to evaluate the N category in gastric cancer; thus, many studies have shown that the LNR is a better predictor of mortality than N category in high-risk or elderly patients who need limited LN dissection [42,45]. ………………………………………..In recurrent thyroid cancer, the LNR may estimate outcomes better than the total number of positive LNs, as the LNR considers the extent of surgery [23]. ……………………………
Tables2 and 3:
The multivariate cox regression model contains variables that ruin the output and should be removed. These very wide confidence interval and poor precision indicates lower insufficient sample size per group.
Response) We sincerely appreciate your comments that make our work improve further. As you advised, all variables with wide confidence intervals were excluded from the tables. Since those variables were not significant in the univariate analysis, they were not previously included in the multivariate analysis and did not change the results.
Thank you.
We thank you and the reviewers for the insightful comments. We believe that our manuscript has been improved as a direct result of the review process. We hope that the revised manuscript is now suitable for publication in CANCERS.
Sincerely,
Joonseon Park, MD
Kwangsoon Kim, MD, PhD

Reviewer 3 Report
Dear authors,
Your proposed manuscript for publication is up to date. For your improvement, I think you should revise the English language. There are minor errors that take away from its flow. I recommend the authors quote the following paper, which contains very interesting information: https://doi.org/10.3390/life12091314. Please add the Ethics Committee Approval number. Please add inclusion/ exclusion criteria. Did you encounter any limitations?
Author Response
Your proposed manuscript for publication is up to date. For your improvement, I think you should revise the English language. There are minor errors that take away from its flow. I recommend the authors quote the following paper, which contains very interesting information: https://doi.org/10.3390/life12091314.
Response) We sincerely appreciate your comments that make our work improve further. As we reviewed the above paper, carcinoma showing thymus-like differentiation (CASTLE) is essentially diagnosed by pathological and immunohistochemical examinations,and positive CD5 immunoreactivity. However, our institution has not routinely checked the CD5 immunoreactivity. We will continue to pay attention to CASTLE and apply it well to our future research.
Please add the Ethics Committee Approval number.
Response) We added the number in “2. Materials and Methods - 2.1. Patients” and at the end of the document:
Institutional Review Board Statement: This study was approved by the Institutional Review Board of Seoul St. Mary’s Hospital, The Catholic University of Korea (IRB No.: KC22RISI0677 and date of approval: 26 September 2022).
Please add inclusion/ exclusion criteria.
Response) Inclusion/ exclusion criteria are described in “2. Materials and Methods - 2.1. Patients”:
Inclusion criteria: Patients with recurrent thyroid cancer who underwent initial thyroid surgery from January 2000 to December 2019 at Seoul St. Mary’s Hospital (Seoul, Korea).
Exclusion criteria: Patients with distant metastasis at the initial operation and the first recurrence were excluded. Ten patients with distant metastasis at the first recurrence were excluded, and 4, 7, and 6 patients were excluded due to transfer, loss to follow-up, and no further treatment (observation only) due to the patient’s decision, respectively.
Did you encounter any limitations?
Response) I described limitations at the end of discussion:
There are several limitations to this study. First, this is a retrospective, single-center study. Second, postoperative Tg values and BIR were not analyzed. Previous studies showed that stimulated Tg is a significant predictor of re-recurrence after the initial re-operation in patients with recurrent PTC [14-16]. Most studies defined BIR as stimulated Tg ≥ 1 ng/mL [14-16], and Lee et al. demonstrated that BIR significantly lowered RFS after 5 years compared to BCR (HR, 3.191; 95% CI, 1.519–6.705; p = 0.001) [14]. In our study, if BIR was suspected, only cases with structural incomplete response diagnosed with additional imaging and pathologic confirmation were included. We will investigate more developed outcomes by including both unstimulated and stimulated postoperative Tg values in the future. Third, although we followed up for an average of 129 months, we found very few cases of DSM. Therefore, a multicenter study with a larger sample size that includes information, such as postoperative Tg values and biochemical responses, could support the reliability of this study.
We thank you and the reviewers for the insightful comments. We believe that our manuscript has been improved as a direct result of the review process. We hope that the revised manuscript is now suitable for publication in CANCERS.
Sincerely,
Joonseon Park, MD
Kwangsoon Kim, MD, PhD

Round 2
Reviewer 2 Report
Authors address all concerns. Thank you.